

# Effectiveness of gamma-oryzanol in glycaemic control and managing oxidative stress, inflammation, and dyslipidaemia in diabetes: a systematic review of preclinical studies

Mustapha Ismail Radda[1,2], Norsuhana Omar[1], Siti Fairuz Mohd Yusof[1], Rozaziana Ahmad[1], Abdul Jalil Rohana[3], Wan Rosli Wan Ishak[4], Anani Aila Mat Zin[5] and Aminah Che Romli[1]

[1] Department of Physiology, School of Medical Sciences, Universiti Sains Malaysia, Kubang Kerian, Kelantan, Malaysia

[2] Department of Physiology, College of Health Sciences, Federal University Dutsin-Ma, Dutsin-Ma, Katsina, Nigeria

[3] Department of Community Medicine, School of Medical Science, Universiti Sains Malaysia, Kubang Kerian, Kelantan, Malaysia

[4] Nutrition and Dietetics Programme, School of Health Sciences, Universiti Sains Malaysia, Kubang Kerian, Kelantan, Malaysia

[5] Department of Pathology, School of Medical Sciences, Universiti Sains Malaysia, Kubang Kerian, Kelantan, Malaysia

Corresponding author
Norsuhana Omar, suhanakk@usm.my

## ABSTRACT

**Background**. Diabetes mellitus (DM) and associated complications remain a global public health challenge despite many confrontational aspects of the disease, and its prevalence is projected to rise in the coming decades. Thus, there is an urgent need to intensify the current efforts to address both the prevalence and adverse effects of diabetes, including the use of natural products. Increasing evidence from the scientific literature has revealed the beneficial effects of gamma oryzanol for treating diabetes and its related complications.

**Aim**. To investigate the effectiveness of gamma oryzanol (γ-oryzanol) in managing hyperglycaemia, oxidative stress, inflammation, and dyslipidaemia in a rodent model of diabetes mellitus.

**Methodology**. The review was conducted by searching PubMed, ScienceDirect, Scopus, and Web of Science for articles published from inception to July 12, 2025, with the terms (Gamma-oryzanol OR γ-oryzanol OR Oryzanol OR Cycloartenyl ferulate OR Gammariza) AND (Diabetes mellitus OR Type 2 diabetes mellitus OR hyperglycemia OR oxidative stress OR inflammation OR dyslipidaemia). The review included only articles that used rat and mouse models of diabetes mellitus and γ-oryzanol as treatments; articles that did not meet these criteria were excluded. A total of nine articles were identified, encompassing a total population of 394 rodents. SyCLE's risk of bias tool was used to assess the methodological quality of the studies.

**Results**. Out of 1,989 records initially identified through the systematic search, nine studies met the eligibility criteria. All included studies were assessed to have an unclear to low risk of bias. The synthesised findings indicate that γ-oryzanol (γ-ORZ) exerts beneficial effects on glycaemic control by enhancing insulin secretion and

sensitivity, as well as by reducing fasting blood glucose (FBG) levels. Additionally, γ-ORZ demonstrates antioxidant activity by elevating endogenous antioxidant enzyme levels and decreasing oxidative stress markers. Its lipid-modulatory effects include the elevation of beneficial lipid fractions and the reduction of atherogenic lipids, thereby alleviating diabetic dyslipidaemia. Moreover, γ-ORZ exhibits anti-inflammatory properties through the downregulation of proinflammatory biomarkers. Despite these promising results in preclinical models, further high-quality investigations, particularly well-designed clinical trials, are essential to validate these findings and support the potential integration of γ-ORZ into diabetes management strategies.

**Conclusion**. Most included studies reported that γ-ORZ positively affected hyperglycaemia, oxidative stress, dyslipidaemia, and inflammation under diabetic conditions. Further research, particularly rigorously designed clinical trials, is strongly recommended to confirm and translate these preclinical findings into clinical practice.

## INTRODUCTION

The global prevalence of DM among adults has risen to 11.1%, with nearly half of the affected individuals unaware of their condition. By 2050, this prevalence is projected to increase to 13% of the global population, representing a 45% rise (*International Diabetes Federation, 2025*). In Malaysia, DM prevalence has grown substantially, from 8.3% in 2015 to 9.4% in 2019, and reached 15.6% by 2023 (*National Health Morbidity Survey, 2023*). This marks an almost twofold increase in less than a decade. Globally, DM is the ninth leading cause of death, responsible for approximately one million deaths each year (*Cooppan, 2016*; *Lin et al., 2020*; *GBD 2021 Diabetes Collaborators, 2023*).

The pathogenesis and progression of DM and its complications are multifactorial, involving complex interactions between genetic susceptibility and environmental influences (*Becker, Simonovich & Phelps, 2019*; *Skyler et al., 2017*). These factors are strongly associated with chronic inflammation (*Banday, Sameer & Nissar, 2020*; *Ohiagu, Chikezie & Chikezie, 2021*; *Zhao et al., 2024*), metabolic dysregulation such as dyslipidaemia and oxidative stress (*Banday, Sameer & Nissar, 2020*; *Guo, Cui & Meng, 2023*; *Schwartz et al., 2017*; *Zhao et al., 2024*), and chronic hyperglycaemia resulting from insulin resistance or deficiency (*Galicia-Garcia et al., 2020*). Given the significant role of oxidative stress, lipid abnormalities, and inflammation in the pathophysiology of DM, there is increasing interest in therapeutic strategies involving antioxidants, lipid-lowering agents, and anti-inflammatory compounds to mitigate the disease and its complications (*Guo, Cui & Meng, 2023*; *Nakamura, 2024*; *Pollack et al., 2016*; *Radda et al., 2025*; *Zhao et al., 2024*).

Various preclinical and clinical studies have reported the antidiabetic properties of numerous plant-derived extracts, including *Cuminum cyminum*, *Urtica dioica*, and *Anacardium occidentale*. However, systematic reviews and meta-analyses have challenged

the consistency and reliability of these claims. For example, a meta-analysis conducted by *Karimian, Farrokhzad & Jalili (2021)* found that *Cuminum cyminum* supplementation did not significantly affect FBG levels or the homeostatic model assessment of insulin resistance (HOMA-IR) in individuals with type 2 diabetes mellitus (T2DM), thereby questioning its clinical utility for glycaemic control. Similarly, a systematic review and meta-analysis by *Jamshidi et al. (2021)*, which examined the effects of *Anacardium occidentale* on glycaemic control and body composition, reported no statistically significant improvements in glycaemic indices or anthropometric parameters. In another comprehensive review, *Tabrizi et al. (2021)* investigated the efficacy of *Urtica dioica* in T2DM management. While modest improvements were observed in FBG, glycated haemoglobin (HbA1c), and triglyceride (TG) levels, the intervention had no significant effect on insulin, total cholesterol (TC), low-density lipoprotein cholesterol (LDL-C), high-density lipoprotein cholesterol (HDL-C), or body mass index (BMI).

In contrast, emerging evidence from a recent narrative review by *Radda et al. (2025)* suggests that γ-ORZ, a bioactive compound derived from brown rice (*Oryza sativa*), holds promising therapeutic potential for the management of T2DM and its associated macrovascular and microvascular complications. These effects include improved glycaemic control (*Alwadani et al., 2022*; *Francisqueti-Ferron et al., 2022*; *Mattei et al., 2021*; *Siqueira et al., 2024*), enhanced pancreatic insulin secretion (*Francisqueti-Ferron et al., 2022*; *Wang et al., 2017*), antioxidative properties (*Alwadani et al., 2022*; *Francisqueti-Ferron et al., 2022*; *Rungratanawanich, Abate & Uberti, 2020*), increased insulin sensitivity, and reduced insulin resistance, which are two critical pathological features of T2DM (*Adamu et al., 2017*; *Francisqueti-Ferron et al., 2022*; *Francisqueti et al., 2017*; *Francisqueti et al., 2018*; *Rungratanawanich, Abate & Uberti, 2020*). Additionally, γ-ORZ has been shown to attenuate dyslipidaemia (*Francisqueti et al., 2017*; *Francisqueti et al., 2018*; *Kobayashi et al., 2019*; *Yan et al., 2022*) and modulate inflammatory pathways (*Francisqueti-Ferron et al., 2022*; *Francisqueti-Ferron et al., 2021a*; *Francisqueti-Ferron et al., 2021b*; *Francisqueti et al., 2018*).

Another recent narrative review by *Palacio, Siqueira & Corrêa (2025)* highlighted the role of γ-ORZ in ameliorating obesity-related metabolic disorders, particularly through improved energy metabolism in skeletal muscle. This finding could have implications for muscle-related pathologies and insulin resistance. However, as these reviews lacked systematic methodology and critical appraisal of the included studies, their conclusions remain limited in strength and generalisability.

Therefore, the present review was designed to systematically identify, evaluate, and synthesise available evidence on the efficacy of γ-ORZ compared with no treatment or placebo in improving glycaemic control, oxidative stress, lipid profiles, and inflammatory biomarkers in rodent models of diabetes mellitus. This work aims to provide a comprehensive and methodologically robust summary of current findings, offering insights into the therapeutic potential of γ-ORZ and laying the foundation for future clinical translation in managing diabetes and its complications.

## METHODS

The review team used the Preferred Reporting Items for Systematic Reviews and Meta-Analyses (PRISMA) Guidelines (*Page et al., 2021*) to design and execute the literature search.

### Registration

The review protocol was registered on the International Prospective Register of Systematic Reviews (PROSPERO) database with identification number CRD42024580576. (https://www.crd.york.ac.uk/PROSPERO/view/CRD42024580576).

### Eligibility criteria

Animal models of type 2 diabetes were selected based on their ability to replicate a stable diabetic phenotype with relevant pathophysiological features. These models included diabetes induced by streptozotocin (STZ) alone, STZ combined with a high-fat diet or nicotinamide, and genetically induced models, which are well-documented for their translational relevance (*Gheibi, Kashfi & Ghasemi, 2017*; *Ighodaro, Adeosun & Akinloye, 2017*). Study inclusion was guided by the Population, Intervention, Comparison, Outcome (PICO) framework (*Davies, 2011*; *Richardson et al., 1995*).

The selected populations comprised three rat strains (Wistar, Sprague–Dawley, and hamster) and three mouse strains (C57BL/6, BALB/c/KOR/Stm Slc-Apoe, and genetically induced ob/ob mice). Studies involving species or strains outside of this list were excluded.

Interventions included either pure γ-oryzanol administered at doses ranging from 20 to 2,000 mg/kg or γ-oryzanol-containing formulations at concentrations between 0.1% and 5% (w/w or w/v). Studies that employed brown rice, whole rice bran oil, or whole brown rice as interventions were excluded. Eligible studies were required to include a comparison between γ-oryzanol-treated diabetic animals and untreated/placebo/vehicle-treated diabetic controls. Comparisons involving standard pharmacological agents or combination therapies were excluded to isolate the specific effects of γ-oryzanol.

Primary outcomes of interest included FBG, glucose tolerance test (GTT), and the HOMA-IR. Secondary outcomes comprised antioxidant markers (superoxide dismutase (SOD), catalase (CAT), and glutathione peroxidase (GPx)); oxidative stress markers (malondialdehyde (MDA), advanced glycation end-products (AGEs), and protein carbonyls (PC)); lipid profile parameters TC, TG, LDL, and HDL); and inflammatory biomarkers, both proinflammatory (interleukin-1β (IL-1β), interleukin-6 (IL-6), and tumour necrosis factor-alpha (TNF-α)) and anti-inflammatory (interleukin-10 (IL-10), interleukin-33 (IL-33), and adiponectin). Studies reporting outcomes outside of these predefined markers were excluded.

### Data sources and search strategy

Two reviewers (MIR and SFMY) independently conducted the literature search and screening process. Discrepancies during the screening phase were resolved through discussion with two additional reviewers (NO and RA). The search strategy was collaboratively developed by the entire review team and subsequently peer-reviewed

by Gambo Umar Danmusa (GUD), an information specialist at the University Library Complex, Federal University Dutsin-Ma, Nigeria. His expert input improved the specificity of the search by minimising the retrieval of irrelevant studies, particularly those focused on metabolic syndrome, and increasing the likelihood of identifying all pertinent publications.

The final search string incorporated the following terms: (Gamma-oryzanol OR γ-oryzanol OR Oryzanol OR Cycloartenyl ferulate OR Gammariza) AND (Diabetes mellitus OR Type 2 diabetes mellitus OR Hyperglycemia OR Oxidative stress OR Inflammation OR Dyslipidaemia). Using this strategy, MIR and SFMY systematically searched four electronic databases: PubMed, Web of Science, Scopus, and ScienceDirect from inception to July 12, 2025, without applying language restrictions. To further ensure comprehensiveness, reference lists of all included studies were manually screened, and both backward and forward citation tracking were performed.

## Study selection and screening

MIR and SFMY independently screened the titles and abstracts of all retrieved records to assess eligibility. ACR was responsible for obtaining the full texts of all potentially eligible articles. MIR and SFMY then independently evaluated the full-text articles for final inclusion. RAJ conducted forward and backward citation analyses to identify additional relevant studies. WRNI and AAMZ engaged in a roundtable discussion to resolve any disagreements during the screening and inclusion process.

## Data extraction

MIR and SFMY independently reviewed the full texts of all eligible studies and extracted relevant data using a standard data extraction form. The extracted information included the following study characteristics: first author, year of publication, sample size, animal species, age, sex, body weight, diabetes induction model, dose of STZ used, diagnostic criteria for diabetes (based on blood glucose levels), γ-ORZ dosage, treatment duration, key findings, and the country in which the study was conducted.

## Assessment of the risk of bias

MIR and NO independently assessed the risk of bias (RoB) for each included study using the Systematic Review Centre for Laboratory Animal Experimentation's (SYRCLE) risk of bias tool (*Hooijmans et al., 2014*). This tool evaluates methodological quality across ten domains: sequence generation, baseline characteristics, appropriate timing of disease induction, and allocation concealment (selection bias); random housing and blinding of the treatment administrator (performance bias); random outcome assessment and blinding of the outcome assessor (detection bias); completeness of outcome data (attrition bias); freedom from selective outcome reporting (reporting bias); and identification of any other potential sources of bias (other bias). Discrepancies in risk assessments were resolved through consultation with RO. Each study was categorised as having a low, high, or unclear risk of bias based on the number of domains judged to fall into each risk category.

## RESULTS

### Study selection

Figure 1 presents the study selection process in accordance with the PRISMA guidelines. The initial database search retrieved 1,989 records. Following deduplication using Mendeley Reference Manager (Mendeley Desktop, version 1.19.8), 23 duplicates were removed. Title and abstract screening resulted in the exclusion of 1,953 irrelevant studies. The remaining 13 articles were assessed in full. Of these, four studies were excluded due to unrelated outcome measures or the use of inappropriate disease models. Consequently, a total of nine studies met the inclusion criteria and were included in the final review.

### Study characteristics

Table 1 summarises the key characteristics of the nine studies included. The earliest study was conducted by *Chen & Cheng (2006)*, while the most recent was by *Bhaskaragoud, Chatterjee & Suresh Kumar (2020)*. Across all studies, the total number of animals was 394, comprising 252 in the experimental groups and 142 in the control groups. Regarding species distribution, 346 were Wistar rats and 48 were mice. Six studies (*Bhaskaragoud et al., 2018*; *Bhaskaragoud, Chatterjee & Suresh Kumar, 2020*; *Chen & Cheng, 2006*; *Cheng et al., 2010*; *Chou et al., 2009*; *Kozuka et al., 2017*) used only male animals, while the remaining three (*Ghatak & Panchal, 2012a*; *Ghatak & Panchal, 2012b*; *Ghatak & Panchal, 2014*) included both sexes.

Regarding the type of DM model, five studies (*Bhaskaragoud, Chatterjee & Suresh Kumar, 2020*; *Chen & Cheng, 2006*; *Cheng et al., 2010*; *Chou et al., 2009*; *Kozuka et al., 2017*) induced T2DM. Among these, two studies (*Bhaskaragoud et al., 2018*; *Bhaskaragoud, Chatterjee & Suresh Kumar, 2020*) employed the HFD/STZ induction method, while the other three (*Chen & Cheng, 2006*; *Cheng et al., 2010*; *Chou et al., 2009*) used the STZ/nicotinamide model. Another three studies (*Ghatak & Panchal, 2012a*; *Ghatak & Panchal, 2012b*; *Ghatak & Panchal, 2014*) induced diabetes using low-dose STZ protocols, and one study (*Kozuka et al., 2017*) employed a genetically induced T2DM model.

Two studies (*Bhaskaragoud et al., 2018*; *Ghatak & Panchal, 2014*) investigated diabetic nephropathy, while one (*Ghatak & Panchal, 2012b*) focused on diabetic neuropathy. The remaining studies used T2DM models without addressing specific diabetic complications. However, none of the included studies followed the widely recommended protocol of inducing obesity through HFD feeding before diabetes induction, a method that more accurately mimics human T2DM pathophysiology, which often involves insulin resistance secondary to obesity. Although T2DM may also develop independently through impaired insulin secretion or in combination with insulin resistance, current experimental guidelines recommend initial induction of insulin resistance *via* HFD, followed by low-dose STZ (20–35 mg/kg, IV or IP) to partially impair pancreatic β-cell function (*Ghasemi & Jeddi, 2023*). In non-HFD-fed models, moderate STZ doses (40–55 mg/kg) may also be used to induce T2DM (*Ghasemi & Jeddi, 2023*). Among the studies using the HFD/STZ model, only *Bhaskaragoud, Chatterjee & Suresh Kumar (2020)* followed this recommended low-dose STZ protocol (30 mg/kg), while *Cheng et al. (2010)* used an intermediate dose (45 mg/kg).

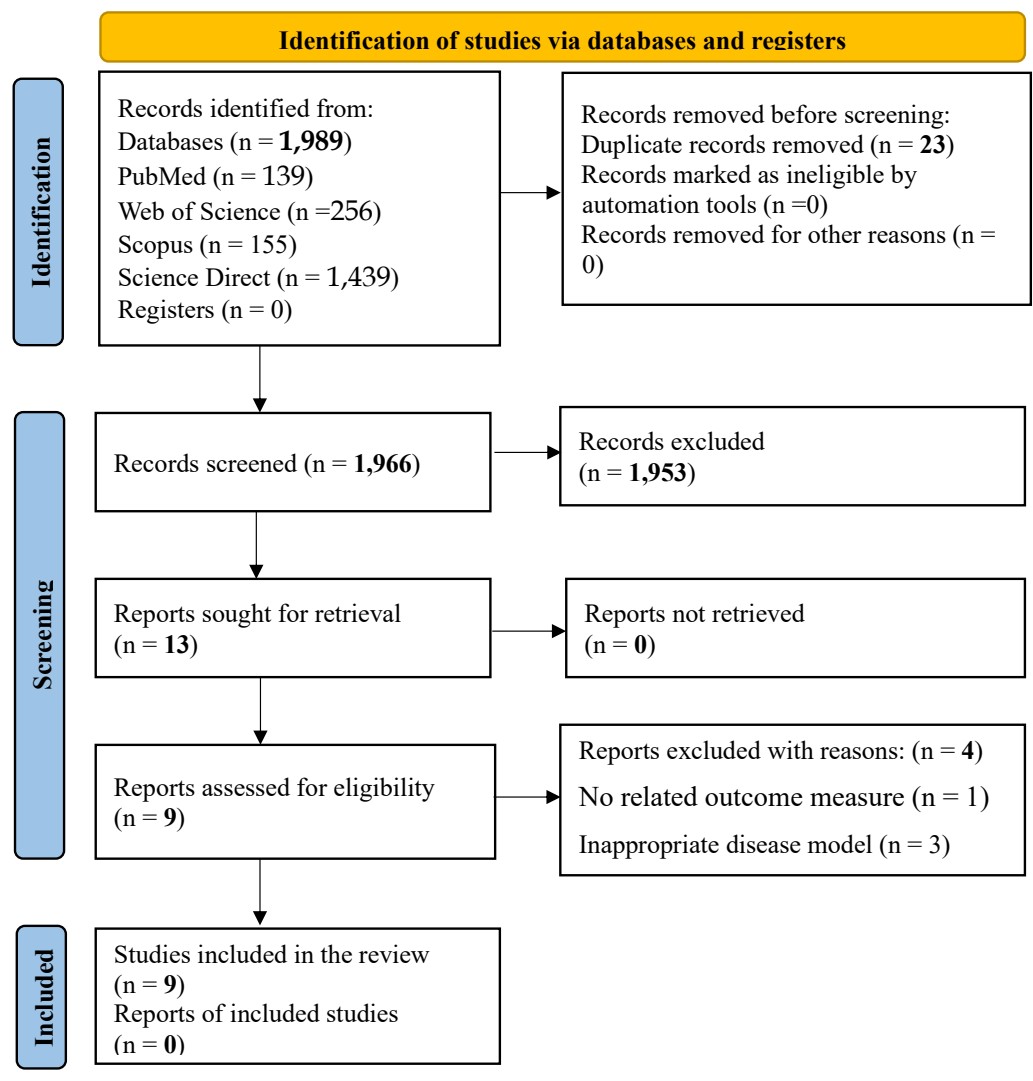

**Figure 1 PRISMA flow chart for the search strategy.** The study selection process for this systematic review followed a structured approach comprising identification, screening, eligibility assessment, and final inclusion. An initial total of 1,989 records were retrieved through database searches, from which 23 duplicates were removed. Title and abstract screening led to the exclusion of 1,953 records. Subsequently, 13 full-text articles were evaluated for eligibility. Ultimately, nine articles (0 reports) met the inclusion criteria, while four were excluded due to either the absence of relevant outcome measures or the use of inappropriate disease models.

Concerning the diagnosis of diabetes, five studies (*Bhaskaragoud et al., 2018*; *Bhaskaragoud, Chatterjee & Suresh Kumar, 2020*; *Chen & Cheng, 2006*; *Cheng et al., 2010*; *Chou et al., 2009*) confirmed hyperglycaemia at least seven days post-STZ injection. Of these, only *Bhaskaragoud et al. (2018)* explicitly reported a diagnostic glucose threshold of >200 mg/dL. Three studies (*Chen & Cheng, 2006*; *Cheng et al., 2010*; *Chou et al., 2009*) used a fasting plasma glucose threshold of >180 mg/dL for diabetes confirmation. Two other studies (*Ghatak & Panchal, 2012b*; *Ghatak & Panchal, 2014*) used a glucose level

>250 mg/dL measured 48 h post-STZ injection, while *Ghatak & Panchal (2012a)* reported a threshold of 300 mg/dL at 72 h post-injection. The genetically induced model (*Kozuka et al., 2017*) did not report a diagnostic criterion. It is important to note that glucose measurements at 48–72 h post-STZ may be prone to misclassification, as some animals, particularly those given lower STZ doses, may exhibit transient hyperglycaemia and return to normoglycaemia within seven days. Therefore, diagnostic confirmation at or beyond day 7 post-STZ is recommended for greater accuracy.

Regarding the mode of γ-ORZ administration, six of the nine studies delivered γ-ORZ mixed with food. However, this approach may reduce intervention fidelity and introduce potential contamination effects or unit-of-analysis errors. The remaining three studies administered γ-ORZ orally as a distinct treatment, thereby avoiding these methodological concerns.

## Assessment of methodological quality

The risk of bias for the included studies is illustrated in Figs. 2 and 3, using the traffic light format (*McGuinness & Higgins, 2021*). None of the studies met the criteria for high methodological quality according to the SYRCLE risk of bias tool. This was primarily due to insufficient reporting across several critical domains. Specifically, all studies exhibited an unclear risk for allocation concealment, random housing, blinding of the intervention administrator, random outcome assessment, and blinding of the outcome assessor (100% unclear risk for each). Moreover, all studies demonstrated a high risk for inadequate sequence generation (100% high risk).

Three studies (*Ghatak & Panchal, 2012a*; *Ghatak & Panchal, 2012b*; *Ghatak & Panchal, 2014*) reported randomisation in group allocations; however, none described the randomisation procedures in detail. These three studies and one additional study (*Kozuka et al., 2017*) were the only ones that avoided unit-of-analysis errors and were thus rated as low risk in the "other bias" domain. Four studies (*Chen & Cheng, 2006*; *Cheng et al., 2010*; *Chou et al., 2009*; *Kozuka et al., 2017*) reported comparable baseline characteristics across study groups, contributing to a lower risk of selection bias in this aspect.

All included studies (100%) were assessed as low risk for selective outcome reporting. Only one study (*Bhaskaragoud, Chatterjee & Suresh Kumar, 2020*) exhibited incomplete outcome data, contributing to an attrition bias risk.

Overall, the methodological quality of the included studies ranged from unclear to low risk of bias, with significant limitations related to the reporting of randomised procedures and blinding strategies.

## Glycaemic control

All nine included studies assessed and reported blood glucose levels as a primary outcome. Of these, six studies (*Bhaskaragoud et al., 2018*; *Bhaskaragoud, Chatterjee & Suresh Kumar, 2020*; *Ghatak & Panchal, 2012a*; *Ghatak & Panchal, 2012b*; *Ghatak & Panchal, 2014*; *Kozuka et al., 2017*) demonstrated that γ-ORZ effectively ameliorated hyperglycaemia in diabetic animal models. Conversely, three studies (*Chen & Cheng, 2006*; *Cheng et al., 2010*; *Chou et al., 2009*) reported no significant glucose-lowering effects following γ-ORZ administration.

Radda et al. (2025), *PeerJ*, DOI 10.7717/peerj.20062

**Table 1 Characteristics of the included studies.** The table summarises key details such as animal characteristics (age, sex & weight), study design, disease model, diagnostic titre, doses of streptozotocin and γ-oryzanol used, treatment duration, and outcome measures evaluated across the included studies.

| Author/Year | Sample size/species | Age/Sex | Body weight | Disease model | Dose of STZ | DM Diagnosis compared with the untreated group | Dose | $R_X$ Duration | Results of the treated group compared with the untreated group | | | | | | Country |
|---|---|---|---|---|---|---|---|---|---|---|---|---|---|---|---|
| | | | | | | | | | Glycaemic parameters | Antioxidants | Prooxidants | Dyslipidaemias | Proinflammatory | Anti-inflammatory | |
| *Chen & Cheng (2006)* | 32 Wistar rats | 7 wks/M | 200 ± 10 g | T2DM | IP 45 mg/kg BW/(200 mg/kg BW nicotinamide | 10 mmol/L 14 days post-STZ injection | 35.2, and 52.8 g γ-ORZ/kg diet | 4 wks | **FBG**: NS **GTT**: NA **HbA1c**: NA **HOMA-IR**: NA | **SOD**: NA **CAT**: NA **GPx**: NA | **MDA**:NA **AGE**: NA **PC**: NA | **TC**: NS ↓↓**TG** **HDL**: NS ↓↓**LDL** | **IL-1β**: NA **IL-6**: NA **TNF-α**: NA | **IL-10**: NA **IL-33**: NA **ApN**: NA | Taiwan |
| *Chou et al. (2009)* | 16 Wistar rats | 6 wks/M | 200 ± 10 g | T2DM | IP 45 mg/kg BW/(200 mg/kg BW nicotinamide | 10 mmol/L 14 days post-STZ injection | 5.25 g γ-ORZ/kg diet | 5 wks | **FBG**: NS **GTT**: NA **HbA1c**: NA **HOMA-IR**: NA | **SOD**: NA **CAT**: NA **GPx**: NA | **MDA**:NA **AGE**: NA **PC**: NA | **TC**: NS **TG**: NS ↑↑**HDL** **LDL**: NS | **IL-1β**: NA **IL-6**: NA **TNF-α**: NA | **IL-10**: NA **IL-33**: NA **ApN**: NA | Taiwan |
| *Cheng et al. (2010)* | 24 Wistar rats | 6 wks/M | 200 ± 10 g | T2DM | IP 45 mg/kg BW/(200 mg/kg BW nicotinamide | 10 mmol/L 14 days post-STZ injection | 5.25 g γ-ORZ/kg diet | 5 wks | **FBG**: NS **GTT**: NA **HbA1c**: NA **HOMA-IR**: NA | **SOD**: NA **CAT**: NA **GPx**: NA | **MDA**:NA **AGE**: NA **PC**: NA | ↓↓**TC** ↓↓ **TG** ↑↑**HDL** ↓↓**LDL** | **IL-1β**: NA **IL-6**: NA **TNF-α**: NA | **IL-10**: NA **IL-33**: NA **ApN**: NA | Taiwan |
| *Ghatak & Panchal (2012a)* | 64 Wistar rats | NR/M&F | 250–300 g | DM neuropathy | IV 45 mg/STZ in citrate buffer (pH 4.5, 0.1 M) was | >250 mg/dL, 48 h post STZ injection | 50 mg/100 mg γ-ORZ | 8 wks | ↓↓**FBG** **GTT**: NA **HbA1c**: NA **HOMA-IR**: NA | **SOD**: NS **CAT**: NS **GPx**: NA | **MDA**: NA **AGE**: NA **PC**: NA | **TC**: NA **TG**: NA **HDL**: NA **LDL**: NA | **IL-1β**: NA **IL-6**: NA **TNF-α**: NA | **IL-10**: NA **IL-33**: NA **ApN**: NA | India |
| *Ghatak & Panchal (2012b)* | 18 Wistar rats | NR/M&F | 250–300 g | T2DM | IV 45 mg/kg of STZ dissolved in citrate buffer (0.1 M, pH 4.5) | >300 mg/dL 72 h post-STZ injection. | 50 mg/100 mg γ-ORZ | 11 days | ↓↓**FBG** **GTT**: NA **HbA1c**: NA **HOMA-IR**: NA | ↑↑**SOD** **CAT**: NA **GPx**: NA | **MDA**: NA **AGE**: NA **PC**: NA | **TC**: NA **TG**: NA **HDL**: NA **LDL**: NA | **IL-1β**: NA **IL-6**: NA **TNF-α**: NA | **IL-10**: NA **IL-33**: NA **ApN**: NA | India |
| *Ghatak & Panchal (2014)* | 64 Wistar rats | NR/M&F | 250–300 g | DM nephropathy | IV 45 mg/kg STZ prepared in citrate buffer (pH 4.5, 0.1 M) | >250 mg/dL, 48 h post STZ injection | 50 mg/100 mg γ-ORZ | 8 wks | ↓↓**FBG** **GTT**: NA **HbA1c**: NA **HOMA-IR**: NA | ↑↑**SOD** ↑↑**CAT** **GPx**: NA | ↓↓**MDA**: **AGE**: NA **PC**: NA | ↓↓**TC** ↓↓ **TG** ↑↑**HDL** ↓↓**LDL** | **IL-1β**: NA **IL-6**: NA **TNF-α**: NA | **IL-10**: NA **IL-33**: NA **ApN**: NA | India |
| *Kozuka et al. (2017)* | 48 Genetically *ob/ob* mice | 5 wks/M | NR | T2DM | Genetically induced *ob/ob* T2DM | NR | 320 mg/g BW γ-ORZ-nanoparticles | 4 wks | ↓↓**FBG** ↓↓**GTT** **HbA1c**: NA **HOMA-IR**: NA | **SOD**: NA **CAT**: NA **GPx**: NA | **MDA**:NA **AGE**: NA **PC**: NA | ↓↓**TC** ↓↓ **TG**: **HDL**: NA ↓↓**LDL** | **IL-1β**: NA ↓↓**IL-6**: ↓↓**TNF-α** | **IL-10**: NA **IL-33**: NA **ApN**: NA | Japan |

| Author/ Year | Sample size/ species | Age/Sex | Body weight | Disease model | Dose of STZ | DM Diagnosis compared with the untreated group | Dose | R$_X$ Duration | Results of the treated group compared with the untreated group | | | | | | Country |
|---|---|---|---|---|---|---|---|---|---|---|---|---|---|---|---|
| | | | | | | | | | Glycaemic parameters | Antioxidants | Prooxidants | Dyslipidaemias | Proinflammatory | Anti-inflammatory | |
| *Bhaskaragoud et al. (2018)* | 64 Wistar rats | NR/M | 100 g | DM nephropathy | IP STZ 30 mg/kg BW | >200 mg/dL, 7 days after STZ injection | 0.1 and 0.3% γ-ORZ concentrate/kg diet | 12 wks | ↓↓**FBG** **GTT:** *NA* **HbA1c:** *NA* **HOMA-IR:** *NA* | ↑↑**SOD** ↑↑**CAT** ↑↑**GPx** | **MDA:** *NA* **AGE:** *NA* **PC:** *NA* | ↓↓**TC** ↓↓**TG** **HDL:** NA **LDL:** NA | **IL-1β:** NA **IL-6:** NA **TNF-α:** NA | **IL-10:** NA **IL-33:** NA **ApN:** NA | India |
| *Bhaskaragoud, Chatterjee & Suresh Kumar (2020)* | 64 Wistar rats | NR/M | 100 g | T2DM | IP STZ 30 mg/kg BW | NR | 0.1 and 0.3% γ-ORZ concentrate/kg diet | 8 wks | ↓↓**FBG** **GTT:** *NS* **HbA1c:** *NA* **HOMA-IR:** *NA* | ↑↑**SOD** ↑↑**CAT** ↑↑**GPx** | ↓↓**MDA** **AGE:** *NA* **PC:** *NA* | ↓↓**TC** ↓↓**TG** **HDL:** NA **LDL:** NA | **IL-1β:** NA **IL-6:** NA **TNF-α:** NA | **IL-10:** NA **IL-33:** NA **ApN:** NA | India |

**Notes.**

Note: DM, Diabetes Mellitus; T2DM, Type-2 Diabetes Mellitus; STZ, Streptozotocin; γ-ORZ, Gamma Oryzanol; Rx, Treatment; RCT, Randomised Controlled Trial; BW, Body Weight; TG, Triglycerides; TC, Total Cholesterol; LDL, Low Density Lipoprotein; HDL, High Density Lipoprotein; SOD, Superoxide Dismutase; CAT, Catalase; GPx, Glutathione Peroxidase; MDA, Malonaldehyde; AGE, Advanced Glycation End-products; PC, Protein Carbonyl; M, Male; F, Female; HFD, High Fat Diet; IP, Intraperitoneal; IV, Intravenous; IL, Interleukin; TNF, Tumour Necrotic Factor; ApN, Adiponectin; FBG, Fasting Blood Glucose; GTT, Glucose Tolerance Test; Hb1AC, Glycated haemoglobin; HOMA-IR, Homeostatic Model Assessment for Insulin Resistance; ↓↓, Significant Decrease; ↑↑, Significant Increase; NS, No Significant effect; NA, Not Assessed.

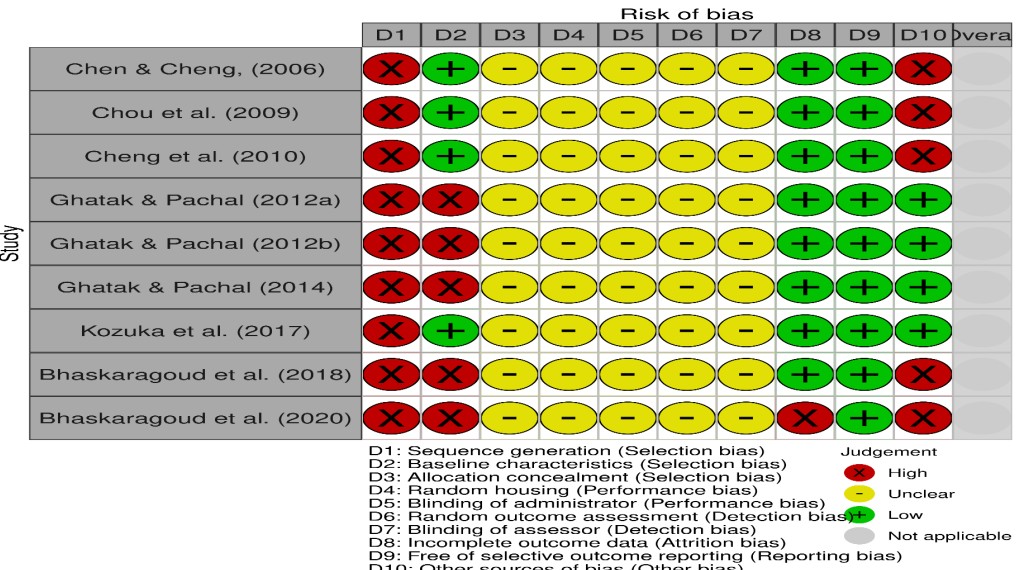

**Figure 2** **Details of the risk of bias judgment per domain for each study analysed using SyCLE's RoB tool.** Each study was evaluated across ten domains: sequence generation, baseline characteristics, allocation concealment, random housing, blinding of carer, random outcome assessment, blinding of assessor, Incomplete outcome data, free of selective outcome reporting, and other sources of bias (other bias). Green (+) indicates low risk of bias, yellow (-) indicates unclear risk, and red (x) indicates high risk. Most studies demonstrated low or unclear risk in most domains, with a few showing high risk, particularly in sequence generation and units of analysis errors. Note: *Chen & Cheng, 2006*; *Chou et al., 2009*; *Cheng et al., 2010*; *Ghatak & Panchal, 2012a*; *Ghatak & Panchal, 2012b*; *Ghatak & Panchal, 2014*; *Kozuka et al., 2017*; *Bhaskaragoud et al., 2018*; *Bhaskaragoud, Chatterjee & Suresh Kumar, 2020*.

Two studies (*Bhaskaragoud, Chatterjee & Suresh Kumar, 2020*; *Kozuka et al., 2017*) evaluated GTT. Among them, only the *Kozuka et al. (2017)* study reported marked improvements in GTT following γ-ORZ treatment. In contrast, *Bhaskaragoud, Chatterjee & Suresh Kumar (2020)* observed no significant effect.

To address the poor intestinal absorption of γ-ORZ, *Kozuka et al. (2017)* compared the efficacy of γ-ORZ-loaded nanoparticles with conventional ORZ. Their findings revealed that the nanoparticle formulation exhibited superior anti-hyperglycaemic efficacy.

Four studies (*Chen & Cheng, 2006*; *Cheng et al., 2010*; *Chou et al., 2009*; *Kozuka et al., 2017*) assessed plasma insulin levels. Of these, *Cheng et al. (2010)* and *Chou et al. (2009)* observed no significant change in insulin concentration but reported notable improvements in insulin sensitivity. In contrast, *Kozuka et al. (2017)* and *Chen & Cheng (2006)* found a significant reduction in circulating insulin levels following γ-ORZ treatment.

Importantly, none of the included studies evaluated long-term glycaemic indicators such as HbA1c or insulin resistance as measured by the HOMA-IR.
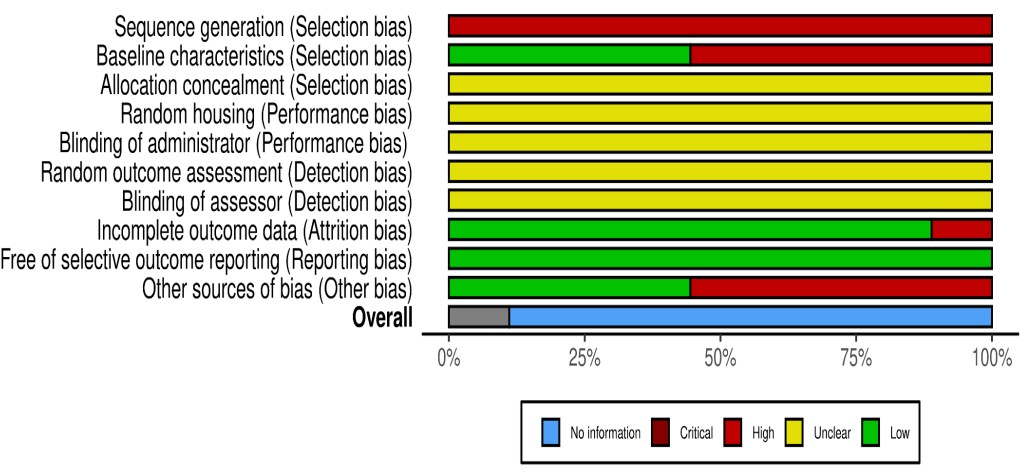

**Figure 3** Summary score of the risk of bias assessment of the included studies based on the SyCLE's RoB tool. The relative distribution of studies showing low (green), unclear (yellow), and high (red) risk of bias for each domain assessed.

## Oxidative stress

Only two studies (*Bhaskaragoud et al., 2018*; *Bhaskaragoud, Chatterjee & Suresh Kumar, 2020*) assessed the three key antioxidant enzymes: SOD, CAT, and GPx, and both reported significant increases in all three markers following γ-ORZ treatment. Two additional studies (*Ghatak & Panchal, 2012b*; *Ghatak & Panchal, 2014*) evaluated the levels of both SOD and CAT. Of these, *Ghatak & Panchal (2014)* observed a marked elevation in both enzymes, while *Ghatak & Panchal (2012b)* reported no significant change. Another study (*Ghatak & Panchal, 2012a*) assessed only SOD and found a notable improvement in its level.

The remaining two studies did not assess any of the three antioxidant enzymes. However, *Kozuka et al. (2017)* examined the impact of γ-ORZ on endoplasmic reticulum (ER) stress-induced pancreatic β-cell apoptosis and found a substantial reduction in β-cell damage, suggesting γ-ORZ may offer cellular protection *via* stress-modulating mechanisms.

Only two studies (*Bhaskaragoud, Chatterjee & Suresh Kumar, 2020*; *Ghatak & Panchal, 2014*) evaluated MDA levels, a key oxidative stress marker, and both reported a significant reduction in MDA concentrations in the γ-ORZ-treated groups. None of the included studies measured other evaluated markers of oxidative damage, AGEs, and PC.

Collectively, these findings suggest that γ-ORZ may alleviate diabetes-induced oxidative stress by upregulating endogenous antioxidant enzymes and reducing oxidative damage, as demonstrated by elevated antioxidant levels and decreased oxidative stress markers.

## Dyslipidaemia

Based on the findings of this review, the mitigation of dyslipidaemia emerged as the most consistently reported beneficial effect of γ-ORZ. All nine included studies (*Bhaskaragoud et al., 2018*; *Bhaskaragoud, Chatterjee & Suresh Kumar, 2020*; *Chen & Cheng, 2006*; *Cheng*

*et al., 2010*; *Chou et al., 2009*; *Ghatak & Panchal, 2014*; *Kozuka et al., 2017*) assessed dyslipidaemia and documented improvements in at least one marker.

Four studies (*Chen & Cheng, 2006*; *Cheng et al., 2010*; *Chou et al., 2009*; *Ghatak & Panchal, 2014*) evaluated all four key lipid profile parameters: TC, TG, HDL, and LDL. Among them, two reported positive effects across all markers, while the other two observed significant improvements in TG, HDL, and LDL but found no impact on TC.

Although HDL was not evaluated, *Kozuka et al. (2017)* assessed TC, TG, and LDL and reported favourable outcomes. The remaining two studies (*Bhaskaragoud et al., 2018*; *Bhaskaragoud, Chatterjee & Suresh Kumar, 2020*) examined only TC and TG and found beneficial effects, with no data reported for HDL or LDL.

These consistent findings across diverse study designs and models suggest that γ-ORZ has a promising lipid-modulating effect, particularly in reducing TG and LDL levels and improving HDL concentrations, thereby potentially mitigating cardiovascular risks associated with diabetes.

### Inflammation

Among all the included studies, only *Kozuka et al. (2017)* evaluated the effect of γ-ORZ on inflammatory markers. The study reported a significant reduction in the levels of two key proinflammatory cytokines, IL-6 and TNF-α, in γ-ORZ-treated animals. However, IL-1β was not assessed. The remaining studies in this review did not investigate any proinflammatory cytokines.

Notably, none of the nine studies evaluated anti-inflammatory cytokines, including IL-10, IL-33, or adiponectin, which limits our understanding of the potential anti-inflammatory mechanisms of γ-ORZ in diabetic models. This gap indicates that future studies must explore both pro- and anti-inflammatory pathways to fully elucidate the immunomodulatory role of γ-ORZ in diabetes.

## DISCUSSION

This systematic review aimed to synthesise existing evidence on the efficacy of γ-ORZ in managing DM and to explore its potential mechanisms of action in mitigating diabetes-related complications. A total of nine preclinical studies were included and qualitatively analysed. Overall findings suggest that γ-ORZ contributes to improved glycaemic control and exerts antioxidative, anti-dyslipidaemic, and anti-inflammatory effects, which may play a role in alleviating the pathophysiological processes associated with DM and its complications.

The therapeutic effects of γ-ORZ are elaborated under the following thematic domains, as illustrated in Fig. 4.

### Glycaemic control

Elevated blood glucose levels, along with physiological alterations, are among the early clinical manifestations of diabetes mellitus (*American Diabetes Association, 2024*). The antihyperglycaemic effects of γ-ORZ may be attributed to several mechanisms. First, γ-ORZ has demonstrated inhibitory activity against α-glucosidase and α-amylase enzymes

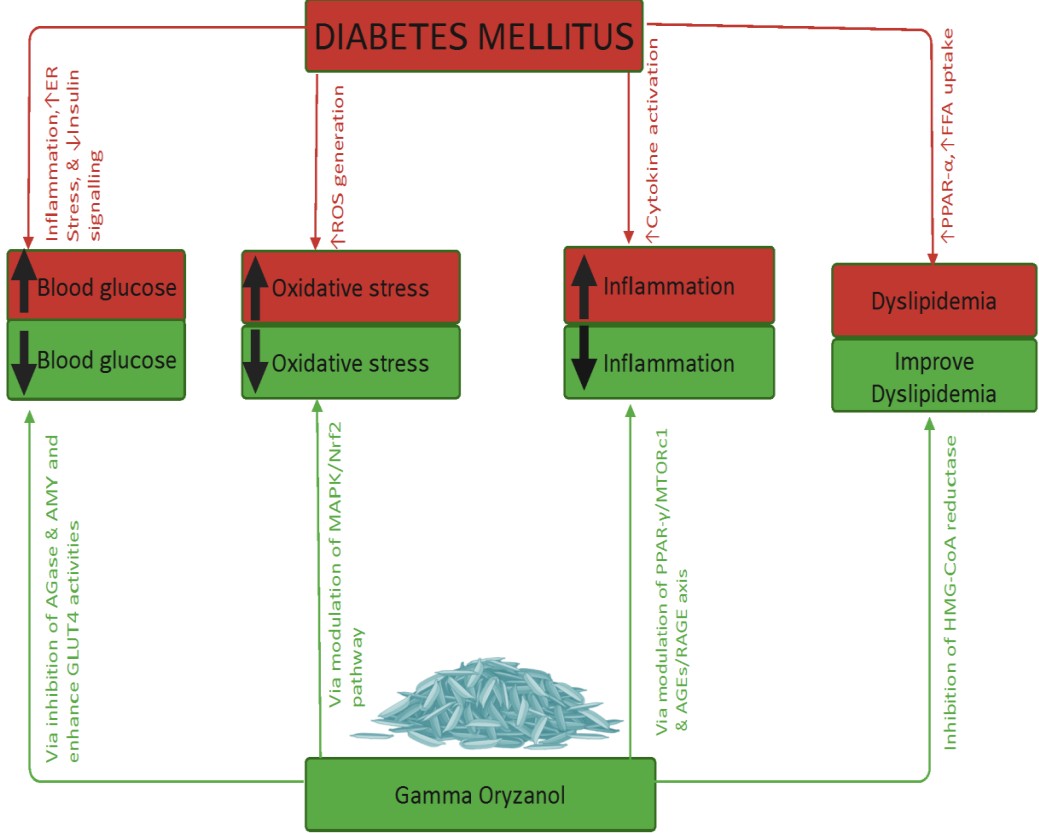

**Figure 4** **Summary findings concerning gamma oryzanol's effects on diabetes and related complications.** Diabetes mellitus results in hyperglycaemia, raised oxidative stress, increased inflammation, and dyslipidaemia. Gamma oryzanol reverses these effects *via* various mechanisms/pathways highlighted above. Note ER, Endoplasmic reticulum; ↓, Decrease; ↑, Increase; ROS, Reactive oxygen species; PPRA, Peroxisome proliferator-activated receptor gamma; FFA, Free fatty acids; AMY, α-Amylase; AGase, α-Glucosidase; GLUT4, Glucose transporter-4; MAPK, Mitogen-Activated Protein Kinase; Nrf2, Nuclear factor erythroid 2-related factor 2; MTORc1, Mechanistic target of rapamycin complex 1; HMG-CoA, Hydroxy-Methyl Glutaryl-Coenzyme A. Created in https://BioRender.com (https://BioRender.com/vbf2i45).

in the gastrointestinal tract, which may reduce carbohydrate digestion and glucose absorption, as shown in an *in vitro* study (*Sansenya, Payaka & Mansalai, 2023*). Second, γ-ORZ may enhance glucose uptake by promoting the translocation of GLUT4 to the cell membrane (*Jung et al., 2015*). Third, it may increase insulin secretion, facilitating glucose utilisation (*Son et al., 2011*), and simultaneously reduce insulin resistance (*Jung et al., 2015*; *Rungratanawanich, Abate & Uberti, 2020*).

Conversely, some studies observed a reduction in circulating insulin levels following γ-ORZ administration. This paradoxical finding could be explained by enhanced insulin sensitivity mediated through upregulation of peroxisome proliferator-activated receptor gamma (PPAR-γ) expression in adipocytes (*Cheng et al., 2010*; *Jung et al., 2015*), which promotes insulin signalling and glucose utilisation in metabolically active tissues.

## Oxidative stress

Oxidative stress arises from an imbalance between prooxidants and antioxidants, and it plays a pivotal role in the development and progression of DM and its complications (*Caturano et al., 2025*). Elevated oxidative stress is consistently linked to the pathophysiology of DM, particularly in exacerbating vascular damage and tissue dysfunction. Numerous rodent studies have demonstrated the therapeutic potential of antioxidants in ameliorating diabetes and its associated complications (*Alqudah et al., 2025*; *Mallik et al., 2024*; *Zhong et al., 2022*), while prooxidants have been shown to contribute significantly to the onset and progression of diabetic pathology (*Choosong et al., 2021*; *Shabalala et al., 2022*; *Shawki et al., 2021*).

In this context, the antioxidative properties of γ-ORZ may involve multiple mechanisms. These include the upregulation of endogenous antioxidants such as SOD, CAT, and GPx, and the reduction of lipid peroxidation, as evidenced by its effect on MDA levels (*Musapoor et al., 2023*). Additionally, γ-ORZ possesses free radical scavenging capacity, likely contributing to its protective effect. γ-ORZ may also exert antioxidant effects by modulating oxidative stress-related signalling pathways, notably the MAPK/Nrf2 pathway, which governs cellular redox homeostasis (*De Gomes et al., 2018*; *Ma et al., 2022*). Collectively, these actions suggest that γ-ORZ can mitigate oxidative stress, thereby potentially delaying the onset of diabetic microvascular and macrovascular complications, and consequently reducing diabetes-associated morbidity and mortality.

## Dyslipidaemia

Dyslipidaemia is a prominent hallmark of DM and a key contributor to the development of diabetes-related complications, particularly cardiovascular diseases (*Lee et al., 2018*; *Liu et al., 2022*; *Lee et al., 2024*; *Pan et al., 2024*; *Wang et al., 2023*). The antihyperlipidaemic effects of γ-ORZ may involve several interrelated mechanisms. One plausible pathway is γ-ORZ's ability to inhibit intestinal cholesterol absorption, likely by competing with cholesterol for incorporation into micelles within the gastrointestinal tract, thereby reducing cholesterol uptake. Additionally, γ-ORZ has been reported to inhibit the activity of 3-hydroxy-3-methylglutaryl-CoA (HMG-CoA) reductase, the rate-limiting enzyme in hepatic cholesterol biosynthesis (*Mäkynen et al., 2012*). Another potential mechanism includes enhancing the faecal excretion of cholesterol and its metabolic by-products, thereby lowering circulating lipid levels (*Srikaeo, 2014*).

## Inflammation

Inflammatory responses play a central role in the pathogenesis and progression of DM and its associated complications (*Zhao et al., 2024*). Inflammation is a contributing factor and can also serve as a biomarker for assessing disease severity and prognosis (*Guo et al., 2022*). The anti-inflammatory effects of γ-ORZ may be attributed to several mechanisms. One proposed mechanism is the modulation of PPAR-γ expression in adipose tissue, which is known to regulate the expression of proinflammatory genes. Another involves suppressing proinflammatory mediator production by peritoneal macrophages, thereby attenuating systemic inflammation (*Francisqueti-Ferron et al., 2021a*; *Francisqueti-Ferron et al., 2021b*).

Additionally, γ-ORZ may exert its anti-inflammatory effects through modulation of the advanced glycation end-products/receptor for advanced glycation end-products (AGE/RAGE) axis, which is closely associated with chronic inflammatory conditions. Its antioxidant properties also enable it to neutralise free radicals, thereby reducing oxidative stress (*Minatel et al., 2016*; *Rao, Sugasini & Lokesh, 2016*). Furthermore, γ-ORZ may attenuate cellular apoptosis (*Huang et al., 2020*) and improve insulin sensitivity (*Rungratanawanich, Abate & Uberti, 2020*), both of which are critical factors in the inflammatory cascade of diabetes.

## Strengths and limitations

To the best of our knowledge, this is the first systematic review to evaluate the efficacy of γ-ORZ in managing hyperglycaemia, oxidative stress, dyslipidaemia, and inflammation in rodent models of DM. The findings provide a foundational understanding of γ-ORZ's potential therapeutic effects in diabetic conditions and related complications, consolidating evidence from multiple preclinical studies. Despite the promising results, the efficacy of γ-ORZ in managing DM and its associated complications remains a subject of ongoing research, primarily due to the limited number of studies available.

All nine studies included in this review were conducted by six research groups from three countries within the same continent, which may limit geographical and methodological diversity. Notably, studies employing two different doses of γ-ORZ (50 mg/kg and 100 mg/kg) reported greater efficacy at the higher dose. Despite an extensive literature search, only nine studies met the inclusion criteria, involving a total of 394 animals, with several studies having sample sizes below 30 animals, highlighting a general limitation in statistical power.

A major limitation of this review is the inability to conduct a meta-analysis due to insufficient data and heterogeneity across studies. Variability in the formulation, dosage, treatment duration, and route of γ-ORZ administration, ranging from *ad libitum* feeding to oral gavage, introduces further complexity. While the *ad libitum* approach may reflect a more natural intake, it raises concerns regarding dose accuracy, especially in sick animals with reduced food or water consumption, and heterogeneous drug distribution. In contrast, oral gavage is recommended for consistent dosing and avoiding unit of analysis errors.

Another limitation is the variable duration of treatment, which ranged from 11 days to 12 weeks, and the use of different γ-ORZ formulations, sometimes mixed with food, further complicating dose estimation and study comparability. Additionally, all included studies exhibited an unclear risk of bias due to inadequate reporting of study protocols, compromising the overall quality and reliability of findings.

Above all, the review is based exclusively on animal data, without including any clinical studies, limiting its translational applicability to human populations. Furthermore, the studies involved multiple animal strains, which may contribute to biological variability. Therefore, the generalisability of the current findings to clinical practice remains uncertain, and caution is warranted in interpreting the results.

## CONCLUSION

This systematic review highlights the preclinical evidence supporting the effectiveness of γ-ORZ in managing DM and mitigating several pathophysiological mechanisms associated with diabetic complications. Overall, the findings suggest that γ-ORZ exhibits promising therapeutic potential, including improved glycaemic control, reduced oxidative stress, modulation of dyslipidaemia, and attenuation of inflammation.

Despite these encouraging outcomes, the current body of evidence is limited to animal studies with relatively small sample sizes and methodological inconsistencies. Therefore, we strongly recommend further high-quality, rigorously designed studies with larger sample sizes, longer treatment durations, and comprehensive assessments of relevant biomarkers. In particular, well-controlled clinical trials are essential to validate these preclinical findings and to assess the safety, efficacy, and translational potential of γ-ORZ for incorporation into diabetes management strategies.

## ACKNOWLEDGEMENTS

We checked the article using a subscribed version of QuillBot in addition to an earlier evaluation using Grammarly.

### Funding

This study was supported by the Fundamental Research Grant Scheme (FRGS) Ministry of Higher Education Malaysia (FRGS/1/2022/SKK10/USM/02/35). There was no additional external funding received for this study. The funders had no role in study design, data collection and analysis, decision to publish, or preparation of the manuscript.

### Grant Disclosures

The following grant information was disclosed by the authors:
The Fundamental Research Grant Scheme (FRGS) Ministry of Higher Education Malaysia: FRGS/1/2022/SKK10/USM/02/35.

### Competing Interests

The authors declare there are no competing interests.

### Author Contributions

- Mustapha Ismail Radda conceived and designed the experiments, performed the experiments, analyzed the data, prepared figures and/or tables, authored or reviewed drafts of the article, and approved the final draft.
- Norsuhana Omar conceived and designed the experiments, performed the experiments, analyzed the data, authored or reviewed drafts of the article, and approved the final draft.
- Siti Fairuz Mohd Yusof conceived and designed the experiments, performed the experiments, prepared figures and/or tables, and approved the final draft.

 

- Rozaziana Ahmad conceived and designed the experiments, analyzed the data, prepared figures and/or tables, and approved the final draft.
- Abdul Jalil Rohana analyzed the data, prepared figures and/or tables, and approved the final draft.
- Wan Rosli Wan Ishak performed the experiments, authored or reviewed drafts of the article, and approved the final draft.
- Anani Aila Mat Zin performed the experiments, authored or reviewed drafts of the article, and approved the final draft.
- Aminah Che Romli performed the experiments, analyzed the data, prepared figures and/or tables, and approved the final draft.

## Data Availability

This is a systematic review/meta-analysis.

## Supplemental Information

Supplemental information for this article can be found online at http://dx.doi.org/10.7717/peerj.20062#supplemental-information.

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
