# Peer review of "Effectiveness of gamma-oryzanol in glycaemic control and managing oxidative stress, inflammation, and dyslipidaemia in diabetes: a systematic review of preclinical studies"

_PeerJ, doi:10.7717/peerj.20062_

## Round 0.1 · original submission · Major Revisions

**Language Note:** The review process has identified that the English language must be improved. PeerJ can provide language editing services - please contact us at [email protected] for pricing (be sure to provide your manuscript number and title). Alternatively, you should make your own arrangements to improve the language quality and provide details in your response letter. – PeerJ Staff

·

Basic reporting

The manuscript reviews the effectiveness of Gamma Oryzanol for managing diabetes and its complications. This is an interesting review but since this is a systematic review of animal studies, it needs to be clearly indicated in the title.
There are also numerous grammatical errors throughout the manuscript that make it difficult to follow

Experimental design

The survey methodology starting from page 91 needs to be revised to improve the understanding of the method used.
The sentence on Line 98-99 appears to be incomplete. Is the sentence on Line 100 a continuation of the previous one? If yes, the organisation of this section is poor and needs to be improved.

Validity of the findings

Figures and Tables are supposed to be self-explanatory. The figure and table legends need improvement to provide more details. Footnotes can also be used to provide more information.

Most of the discussion was too descriptive and not critical enough. Similarly, there is a need to discuss some obvious limitations like the review using preclinical data only, study duration, oryzanol formulations used in the different studies, and issues with generalizability and bias.

Reviewer 2 ·

Basic reporting

I read the manuscript by Radda and colleagues with interest. The authors undertook a systematic review of animal studies, which is commendable. Of particular interest is that the study is registered with PROSPERO. The results are interesting and may contribute meaningfully to the existing body of knowledge.

Experimental design

1) The methodology lacks detail on the use of PICO, PRISMA, and Cochrane guidelines, which are only partially reflected in the results. These should be included in the methods section along with appropriate references.
2) In line 102, the authors mention inclusion of rodent models induced with STZ. Why were other models (e.g., diet-induced or alloxan-induced) excluded? Please justify this choice.
3) In the outcomes section (lines 114–119), the exclusion of studies reporting HOMA-IR and glucose tolerance tests should be justified. These are widely accepted indicators of glycemic control, and their exclusion represents a significant limitation.
4) Focusing only on MDA and SOD is not sufficiently motivated. Why were markers like GSH, GPx, and ROS excluded?
5) Line 121: Was the search conducted independently by MIR and RA, or did they work together? Please clarify for transparency.
6) For inflammation, why were only pro-inflammatory markers considered? What about anti-inflammatory markers? Can pro-inflammatory markers alone fully reflect the inflammatory status?
7) For lipid profiles, only TG and HDL were considered. Why were TC and LDL excluded? These are essential for a complete assessment of dyslipidemia. A clearer definition and understanding of dyslipidemia is needed to ensure accurate conclusions.
8) Line 123: Please specify the name and role of the specialist involved in the literature search.
9) Line 128: There is inconsistency in the reported search date—July 31st in the abstract versus July 28th in the methods. Please correct this to avoid confusion and ensure reproducibility.
10) Line 132: Restricting inclusion to English-only publications may introduce language bias. Please comment on this limitation.
11) Lines 136–140: The link provided redirects to the general Web of Science homepage, not to the exact search strategy. I suggest including the complete search strategy in a supplementary file, including all search terms, restrictions, and number of articles retrieved per database.
12) Line 143: Clarify which authors were involved in specific steps of the process.
13) Line 147: Figure placement may be better suited to the results section rather than the methods.
14) Line 173: Please specify the version of Mendeley used for reference management.
15) Line 187: There is a contradiction. The results mention inclusion of high-fat diet models, but the methods specify only STZ-induced models were included. Please correct this inconsistency.

Validity of the findings

However, the manuscript's quality could be improved by addressing the following points during revision:
1. The abstract and inclusion criteria focus on preclinical studies. I suggest the title reflect this more explicitly, e.g., “Systematic Review of Preclinical Studies.”
2. The background section in the abstract is too detailed. Consider removing the aim from line 26–27, as it duplicates the aim already stated in lines 29–31.
3. In line 30, please insert the abbreviation for gamma oryzanol to aid reader recall as they progress through the manuscript.
4. In line 34, specify the exact dates of the literature search. There is a concern that studies published after July 2025 may not have been included, especially given the growing interest in this topic. I recommend updating the search to ensure inclusion of the most recent literature.
5. Line 41: The term “recruited” may not be appropriate since participants were not recruited. A better term would be “collected” or “identified” from databases.
6. The results section (lines 46–49) should be revised. Consider presenting each outcome in a separate sentence. For example: “Gamma oryzanol reduces hyperglycemia by increasing insulin secretion and sensitivity, along with a reduction in fasting plasma glucose.” Use a similar format for inflammation, oxidative stress, and dyslipidemia.
7. The conclusion should clearly state the specific markers assessed in the study.
8. The keywords should include "hyperglycemia" for better discoverability.
9. The introduction would benefit from citing other studies that have explored herbal therapies in animal models of diabetes. Highlighting their limitations would justify the current study's focus on gamma oryzanol.
10. The discussion is written more like a results section. I suggest separating the results and discussion into distinct sections.
11. In the results, the reader expects to see the effect of ORZ on markers of hyperglycemia, inflammation, oxidative stress, and dyslipidemia. Where possible, include summary data from the original studies reviewed.
12. The discussion should interpret and explain the results — i.e., tell the story. For instance, why did gamma oryzanol reduce hyperglycemia or oxidative stress? What are its proposed mechanisms of action? Which pathways are involved?
13. Line 251: Since GTT was already defined in the methods, use the abbreviation here instead of the full name.
14. Line 256: Define PPAR-γ (peroxisome proliferator-activated receptor gamma) in full on first use.
15. The limitations section needs to be more aligned with reviewer observations. Notably, the small sample size (only 7 studies included), use of different rodent strains (e.g., Sprague-Dawley, ob/ob mice), variation in sample size, treatment duration, and dose regimens should be discussed. Also mention the lack of clinical studies.
16. Line 321: This sentence appears disconnected from the rest of the text. Consider integrating the related figure into the discussion for better flow.
17. The conclusion is too broad. Refine it to summarize key findings and provide a clear take-home message.
18. Figure 5 shows that GOZ reduces blood glucose, oxidative stress, and inflammation, and improves lipid profiles. However, it lacks detail on the mechanisms by which GOZ achieves these effects. Consider including a schematic of the mechanistic pathways.
19. In Table 1, place the treatment duration in a column after the dose. As it stands, the formatting makes it unclear whether the duration refers to the STZ induction or the treatment period.

---

## Round 0.2 · Minor Revisions

Dear authors,

Our reviewers appreciate your sincere efforts to improve the manuscript. However, it still needs some improvement. Please revise and resubmit asap.
All the best.

**Language Note:** The review process has identified that the English language must be improved. PeerJ can provide language editing services - please contact us at [email protected] for pricing (be sure to provide your manuscript number and title). Alternatively, you should make your own arrangements to improve the language quality and provide details in your response letter. – PeerJ Staff

·

Basic reporting

The manuscript has been improved to an acceptable level. The title is clearer, and it has undergone substantial grammatical corrections.

Experimental design

The methodology section has also been improved with more coherent sentence transitions and flow.

Validity of the findings

This is also satisfactory.

Additional comments

There are still some grammatical errors that need to be corrected.

Reviewer 2 ·

Basic reporting

addressed previous comments adequately

Experimental design

-

Validity of the findings

-

---

## Round 0.3 · accepted · Accept

Dear Authors,

On behalf of the editorial board of PeerJ, it is my pleasure to inform that your manuscript was appreciated by all the reviewers and does not need any further changes. Therefore, the manuscript is accepted for publication in PeerJ. Please remember, this is editorial acceptance and still needs certain tasks to be completed before publication. I request you to be available for a few days to avoid any delays.

All the best for your future submissions.

·

Basic reporting

no comment

Experimental design

no comment

Validity of the findings

no comment

Additional comments

The manuscript has been improved and is now acceptable for publication.